# *tp53* deficiency causes a wide tumor spectrum and increases embryonal rhabdomyosarcoma metastasis in zebrafish

Myron S Ignatius[1,2,3,4†], Madeline N Hayes[1,2,3†], Finola E Moore[1,2,3], Qin Tang[1,2,3], Sara P Garcia[1], Patrick R Blackburn[5], Kunal Baxi[4], Long Wang[4], Alexander Jin[1], Ashwin Ramakrishnan[1], Sophia Reeder[1], Yidong Chen[4], Gunnlaugur Petur Nielsen[1,2], Eleanor Y Chen[6], Robert P Hasserjian[1,2], Franck Tirode[7], Stephen C Ekker[8], David M Langenau[1,2,3]*

[1]Department of Pathology, Massachusetts General Hospital Research Institute, Boston, Massachusetts; [2]Center of Cancer Research, Massachusetts General Hospital Cancer Center, Charlestown, Massachusetts; [3]Harvard Stem Cell Institute, Boston, Massachusetts; [4]Department of Molecular Medicine, Greehey Children's Cancer Research Institute, San Antonio, Texas; [5]Department of Laboratory Medicine and Pathology, Mayo Clinic, Rochester, United States; [6]Department of Pathology, University of Washington, Seattle, United States; [7]Department of Translational Research and Innovation, Université Claude Bernard Lyon, Cancer Research Center of Lyon, Lyon, France; [8]Department of Biochemistry and Molecular Biology, Mayo Clinic, Rochester, United States

*For correspondence:
dlangenau@mgh.harvard.edu

†These authors contributed equally to this work

Competing interests: The authors declare that no competing interests exist.

**Abstract** The *TP53* tumor-suppressor gene is mutated in >50% of human tumors and Li-Fraumeni patients with germ line inactivation are predisposed to developing cancer. Here, we generated *tp53* deleted zebrafish that spontaneously develop malignant peripheral nerve-sheath tumors, angiosarcomas, germ cell tumors, and an aggressive Natural Killer cell-like leukemia for which no animal model has been developed. Because the tp53 deletion was generated in syngeneic zebrafish, engraftment of fluorescent-labeled tumors could be dynamically visualized over time. Importantly, engrafted tumors shared gene expression signatures with predicted cells of origin in human tissue. Finally, we showed that tp53$^{del/del}$ enhanced invasion and metastasis in kRAS$^{G12D}$-induced embryonal rhabdomyosarcoma (ERMS), but did not alter the overall frequency of cancer stem cells, suggesting novel pro-metastatic roles for TP53 loss-of-function in human muscle tumors. In summary, we have developed a Li-Fraumeni zebrafish model that is amenable to large-scale transplantation and direct visualization of tumor growth in live animals.
DOI: https://doi.org/10.7554/eLife.37202.001

## Introduction

TP53 is a tumor suppressor protein that is mutated or functionally disrupted in more than 50% of all human tumors (*Kastenhuber and Lowe, 2017*; *Muller and Vousden, 2014*). Moreover, genetic mutation of *TP53* in Li-Fraumeni patients leads to cancer predisposition early in life and is associated with transformation in a broad range of target tissues (*Malkin, 2011*). *TP53* is commonly inactivated by single amino acid mutations that create dominant-negative forms of the protein that inhibit efficient tetramer formation and block transcriptional activity (*Muller and Vousden, 2014*). In this

setting, TP53 alleles likely alter transcriptional activity of TP53 and its related transcription factor family members, TP63 and TP73 (Lang et al., 2004; Olive et al., 2004). By contrast, TP53 deletion is expected to have less wide-ranging transcriptional effects that are confined to tetrameric transcription factor function. Regardless of the genetic alteration, TP53 transcriptional inactivation can lead to genomic instability and impaired apoptotic responses that often are predisposing to a wide array of cancers (Kastenhuber and Lowe, 2017; Muller and Vousden, 2014).

To date, several murine genetic models have been developed to assess the effects of both loss- and gain-of-function Tp53 mutations in cancer (Donehower et al., 1992; Harvey et al., 1993; Jacks et al., 1994; Lang et al., 2004; Lavigueur et al., 1989; Lee et al., 1994; Olive et al., 2004). Both Tp53 mutant and null alleles spontaneously develop cancer. However, similar to human Li-Fraumeni patients, the spectrum varies among different alleles, suggesting that the mode of Tp53 inactivation has important implications in regulating the types of cancer that develop, the time to onset, and the overall propensity for tumor progression (Lavigueur et al., 1989; Lee et al., 1994). For example, mice heterozygous for the 172His point mutation are predisposed to developing osteosarcoma while animals harboring the 270His mutation develop hemangiosarcoma and carcinoma (Olive et al., 2004). By contrast, mice with homozygous Tp53 deletion mainly develop lymphoma, with rare cases of angiosarcoma, undifferentiated sarcoma, osteosarcoma, rhabdomyosarcoma, testicular tumors, nervous system tumors, teratoma, and mammary carcinoma being reported (Donehower et al., 1992; Harvey et al., 1993; Jacks et al., 1994). Together, these data suggest that differences in gain- and loss-of-function alleles have profound effects on tumor onset and spectrum in genetically engineered mice and yet, largely recapitulate the wide array of cancers observed in Li-Fraumeni patients. Importantly, a small subset of Li-Fraumeni syndrome patients harbor genomic deletions in the TP53 locus and cancers that develop in dominant-negative, heterozygous point-mutation carriers often display deletion of the second TP53 allele (Malkin, 2011). Thus, modeling complete TP53 loss-of-function in different animal models will likely provide novel insights into human disease.

TP53 is also commonly mutated in human sarcomas and is predictive of poor outcome (Taubert et al., 1996). For example, the TP53 locus is mutated in 16% of human embryonal rhabdomyosarcoma (ERMS), a common pediatric cancer of muscle and transcriptional activity is altered in >30% of human ERMS through TP53 locus disruption or MDM2 amplification (Taylor et al., 2000). Interestingly, TP53 mutations are also acquired at ERMS relapse (Chen et al., 2013), suggesting a likely role for TP53 in ERMS progression and therapy resistance. Finally, Li-Fraumeni patients with germline TP53 mutations commonly develop ERMS (Malkin, 2011), suggesting important roles for TP53 loss in the genesis of this disease. Yet, to date, the effect of TP53 pathway inactivation on cancer stem cell number, tumor progression, and metastasis in ERMS is not fully understood. Moreover, because genetically engineered mouse and human xenograft models of ERMS do not metastasize in vivo, assessing TP53 loss-of-function in the context of rhabdomyosarcoma metastasis has not been possible. Finally, to date, no tp53 deletion models have been generated in syngeneic zebrafish, precluding large-scale transplantation studies to assess how deletion regulates cancer stem cells and tumor invasion in vivo for a wide array of cancers.

To better study tp53 biology in vivo, we generated a complete loss-of-function tp53 deletion allele in syngeneic CG1-strain zebrafish using TALEN endonucleases. $tp53^{del/del}$ animals spontaneously developed a wide range of tumors including malignant peripheral nerve-sheath tumors (MPNSTs), angiosarcomas, germ cell tumors, and an aggressive natural killer cell-like leukemia not previously described in any animal model. This model contrasts with currently available point-mutation alleles for zebrafish tp53 that predominantly develop MPNSTs (Berghmans et al., 2005). Moreover, because the $tp53^{del/del}$ mutant was generated in CG1-strain syngeneic zebrafish (Mizgireuv and Revskoy, 2006), tumors efficiently transplanted into recipient fish enabling expansion of unlabeled and GFP-labeled tumors, dynamic live animal imaging of metastatic progression, and analysis of transcriptional differences between tumors using RNA sequencing approaches. Roles for tp53 were also assessed in $kRAS^{G12D}$-induced ERMS using large-scale cell transplantation assays and live fluorescent imaging over time. Using these approaches, we showed that the overall frequency of ERMS self-renewing cancer stem cells was unaffected by loss of tp53. In contrast, $tp53^{del/del}$ ERMS were more invasive, providing a potential explanation for increased aggression associated with TP53 disruption in human ERMS (Seki et al., 2015). Taken together, our work has uncovered

novel roles for Tp53 loss in the onset of a wide array of cancers and has provided new insights into how *tp53* affects ERMS progression in vivo.

## Results and discussion

### *tp53* deletion mutants spontaneously develop MPNSTs, angiosarcoma, germ cell tumors, and leukemia

Given the critical function of Tp53 as a tumor suppressor and the absence of a complete null allele in zebrafish, we created a *tp53* deletion mutant using two pairs of TALENs (Transcription Activator-Like Effector Nucleases) that cleaved at the 5' and 3' end of the *tp53* locus (**Figure 1A**). One-cell stage CG1 syngeneic embryos (**Mizgireuv and Revskoy, 2006**) were microinjected with mRNA encoding each TALEN pair and raised to adulthood. F1 embryos were screened by genomic PCR to identify a single founder line with deletion of 12.1 kb *tp53* genomic sequence (**Figure 1A**). CG1 *tp53^wt/del^* heterozygous fish were in-crossed and progeny assessed for Tp53 protein loss and the ability to undergo apoptosis following ionizing irradiation. As expected, homozygous *tp53^del/del^* embryos lacked protein expression and were resistant to radiation-induced apoptosis (**Figure 1—figure supplement 1B–D**).

Progeny from *tp53^wt/del^* crosses were also raised to adulthood and assessed for viability and tumor onset over time. Both heterozygous *tp53 ^wt/del^* and homozygous *tp53^del/del^* fish survived until adulthood at expected ratios (**Figure 1—figure supplement 1A**). By 4 months of age, *tp53^del/del^* zebrafish began to spontaneously develop tumors. Phenotypes of the earliest malignant *tp53^del/del^* cohort were consistent with loss of osmoregulation and kidney damage. Histopathological analysis of these animals revealed features consistent with leukemia, including blast-like cells predominating in the kidney marrow and loss of kidney stromal architecture, including effacement of the renal tubules (**Figure 1B–D**, **Figure 1—figure supplement 2**). Beginning by 7 months of age, a subset of *tp53^del/del^* animals developed externally visible tumors and histology consistent with angiosarcoma and malignant peripheral nerve sheath tumors (MPNSTs) (**Figure 1E–J**, **Figure 1—figure supplement 2**) (**Berghmans et al., 2005**; **Choorapoikayil et al., 2012**; **Parant et al., 2010**). A small subset of *tp53^del/del^* fish also developed prominent externally visible abdominal masses that were diagnosed as germ cell tumors following histopathological analysis (n = 2, **Figure 1K–M**, **Figure 1—figure supplement 2**) (**Neumann et al., 2011**). MPNST assignment was validated using IHC staining for *sox10*, which is a well-established clinical marker for this tumor type (**Figure 1J**) (**Shin et al., 2012**). As expected, *tp53^wt/del^* zebrafish infrequently developed tumors, which is consistent with studies in other *tp53* deficiency mouse models (**Donehower et al., 1992**; **Harvey et al., 1993**; **Jacks et al., 1994**).

In total, 37% of *tp53^del/del^* animals developed externally visible tumors by 12 months of age with a wider tumor spectrum than previously reported in homozygous *tp53^M214K^* and *tp53^I166T^* mutant zebrafish (**Figure 1N,O**) (**Berghmans et al., 2005**; **Parant et al., 2010**). For example, these point mutation models predominantly developed MPNSTs with only a rare, single melanoma being detected in homozygous animals (**Berghmans et al., 2005**). Remarkably, the spectrum in *tp53^del/del^* zebrafish was more similar to that reported in *Tp53*-null mice, with angiosarcomas and germ cell tumors occurring in both models (**Donehower et al., 1992**; **Harvey et al., 1993**; **Jacks et al., 1994**). However, the predominance of T cell lymphomas seen in *Tp53*-null mice was not observed in *tp53^del/del^* zebrafish, likely reflecting species differences in sensitivity to Tp53 loss in target cells. It is also possible that tumor types seen in *Tp53*-null mice exhibit longer latency in our model and would not manifest in the short-lived CG1 strain zebrafish. Finally, we also generated ERMS in *tp53^del/del^* fish by microinjecting linearized human *kRASG12D* oncogene and *GFP* under the control of *rag2* promoter (**Figure 1P,Q**) (**Langenau et al., 2007**). Histopathological analysis of transgene-induced *tp53^del/del^* ERMS showed consistent morphology with the spindle-variant of human ERMS (**Figure 1R,S**) (**Langenau et al., 2007**).

### *tp53^del/del^* tumors are transplantable

One major advantage of generating *tp53^del/del^* mutations in the CG1 syngeneic stain of zebrafish is the ease with which tumors can be used in cell transplantation assays. Primary MPNSTs that arose in the eye were dissected from euthanized *tp53^del/del^* animals and orthotopically transplanted into the

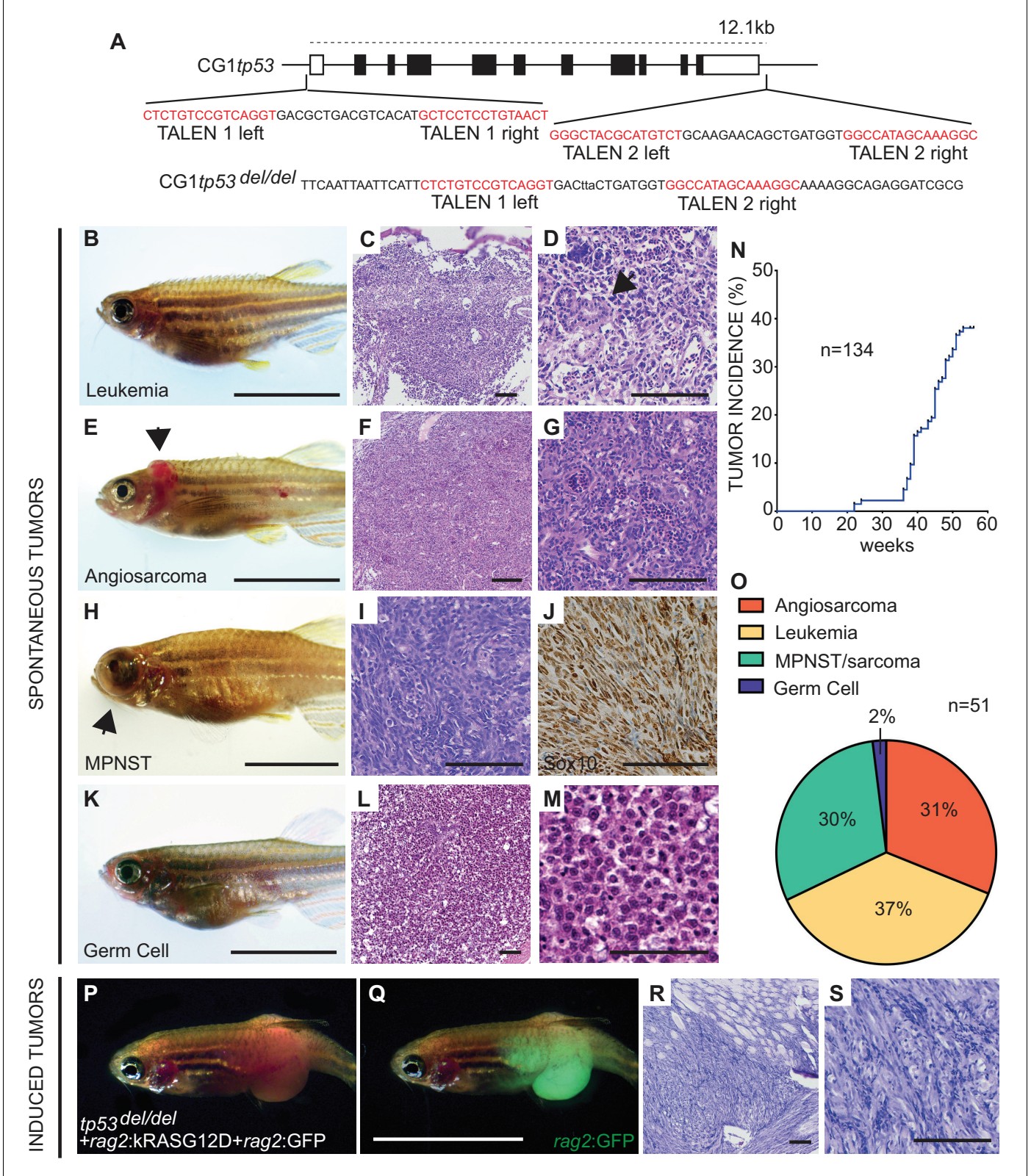

**Figure 1.** Homozygous *tp53*$^{del/del}$ zebrafish spontaneously develop a wide range of tumor types. (**A**) *tp53* genomic locus and CG1 *tp53*$^{del/del}$ allele. TALEN arms were designed to target the 5' and 3' genomic sequence of *tp53* (red). (**B–M**) CG1 *tp53*$^{del/del}$ zebrafish develop leukemia (**B–D**), angiosarcoma (**E–G**), MPNSTs (**H–J**), and germ cell tumors (**K–M**). Whole animal images (**B,E,H,K**), hematoxylin/eosin (H and E) stained sections (**C,D,F, G,I,L,M**), and immunohistochemistry for Sox10 (**J**). Blast-like leukemia cells predominate in the kidney marrow and efface the renal tubules (black arrow,

*Figure 1 continued on next page*

Figure 1 continued

(D). (N) Tumor incidence in CG1 *tp53*<sup>del/del</sup> zebrafish (n = 134). (O) Quantitation of tumor types that form in CG1 *tp53*<sup>del/del</sup> mutant zebrafish by 55 weeks of life based on histology review (n = 51). (P–S) *kRAS*<sup>G12D</sup>-induced embryonal rhabdomyosarcoma (ERMS) generated in CG1 *tp53*<sup>del/del</sup> zebrafish. Whole animal bright field and GFP-epifluorescence overlap images (P and Q, respectively). H and E stained sections revealed features consistent with human ERMS (R,S). Scale bars equal 12.5 mm in whole animal images and 100 μm in histology images.

DOI: https://doi.org/10.7554/eLife.37202.002

The following source data and figure supplements are available for figure 1:

Figure supplement 1. *tp53*<sup>del/del</sup> zebrafish survive at expected Mendelian ratios, lack Tp53 protein expression, and are resistant to irradiation-induced cell death.
DOI: https://doi.org/10.7554/eLife.37202.003
Figure supplement 2. Tumor latency in *tp53*<sup>del/del</sup> zebrafish.
DOI: https://doi.org/10.7554/eLife.37202.004
Figure supplemenrt 2—source data 1. Source data for *Figure 1—figure supplement 2*.
DOI: https://doi.org/10.7554/eLife.37202.005

equivalent periocular space or into the peritoneum of CG1-strain recipient fish (n = 2 primary tumors, n = 9 recipient fish total, $2 \times 10^4$ cells/fish). All recipient animals engrafted tumor with histology similar to the primary disease (*Figure 2A–E*, *Figure 2—figure supplement 1*). To more easily track tumor cells in host animals, we also crossed *tp53*<sup>del/del</sup> animals into CG1 syngeneic *ubi*:GFP transgenic zebrafish. We successfully engrafted *tp53*<sup>del/del</sup> angiosarcomas into CG1-recipient animals and *ubi*:GFP+ tumor cells were easily traceable in non-fluorescent recipients (n = 3 primary tumors, n = 8 of 9 transplant fish developed tumors, *Figure 2F–I*, *Figure 2—figure supplement 1*). Finally, *ubi*:GFP+ leukemias were also assessed by cell transplantation. Specifically, blood cells were engrafted into non-irradiated CG1 strain fish ($2.5 \times 10^4$ cells/intraperitoneal injection). In total, five of five primary leukemias engrafted into recipient fish with GFP+ cells disseminating widely throughout the animal by 60 days post-transplantation (n = 5 primary leukemias, n = 25 of 25 animals engrafted leukemia, *Figure 2J–M*, *Figure 2—figure supplement 1*). Whole animal imaging and flow cytometric analysis revealed that GFP+ cells also invaded the recipient kidney marrow, the site of hematopoiesis in adult zebrafish (*Figure 2M,N*). Leukemic cells consisted of as much as 45% of the reconstituted marrow in transplanted fish (*Figure 2N*, n = 3 independent primary leukemias analyzed). To more closely observe leukemia cell morphology, FACs sorted GFP+ leukemia cells were assessed by cytospin and Wright/Giemsa staining (*Figure 2O–R*). The leukemic cells were large with prominent nucleoli and abundant, vacuolated cytoplasm, consistent with a rare, high-grade aggressive NK cell leukemia. In the context of gene expression profiling (outlined below), these leukemias were also similarly classified as aggressive NK cell-like leukemia, suggesting important roles for *Tp53* in initiation of leukemias of NK cell origin (*Figure 2Q,R*). Finally, GFP-labeled *kRASG12D*-induced *tp53*<sup>del/del</sup> ERMS were readily transplantable when engrafted into syngeneic recipient fish (n = 11 primary tumors analyzed, n = 47/49 fish engrafted, *Figure 2S–V*, *Figure 2—figure supplement 1*). Histology was similar between primary and transplanted MPNSTs, angiosarcomas, leukemias and ERMS (*Figure 2B,C,E,I,M,V*).

## Gene expression analysis of *tp53*<sup>del/del</sup> tumors arising in transplant recipient fish

We next profiled the transcriptome of tumor cells isolated from fish transplanted with *tp53*<sup>del/del</sup> MPNSTs, angiosarcomas, leukemias, a germ cell tumor and *kRAS*<sup>G12D</sup>-induced ERMS by Poly(A)<sup>+</sup> RNA-sequencing (RNAseq). MPNSTs and the germ cell tumor were analyzed from bulk tumor isolated from non-GFP labeled animals, whereas angiosarcomas, leukemias, and ERMS were FACS sorted from engrafted GFP-labeled tumors (purity and viability >85%). Bulk mRNA from three independent wild-type CG1 strain fish was also sequenced and used as a control. Principal component analysis identified six distinct clusters corresponding to whole CG1 syngeneic fish, leukemia, MPNSTs, angiosarcomas, germ cell tumor and ERMS (*Figure 3A*). By comparing gene expression among different tumor types, unique tumor-specific expression profiles were identified and each assessed for overlap with gene sets found in the Molecular Signatures Database (MSigDB, *Figure 3B,C*, *Figure 3—source datas 1–3*). For example, the upregulated leukemia gene set

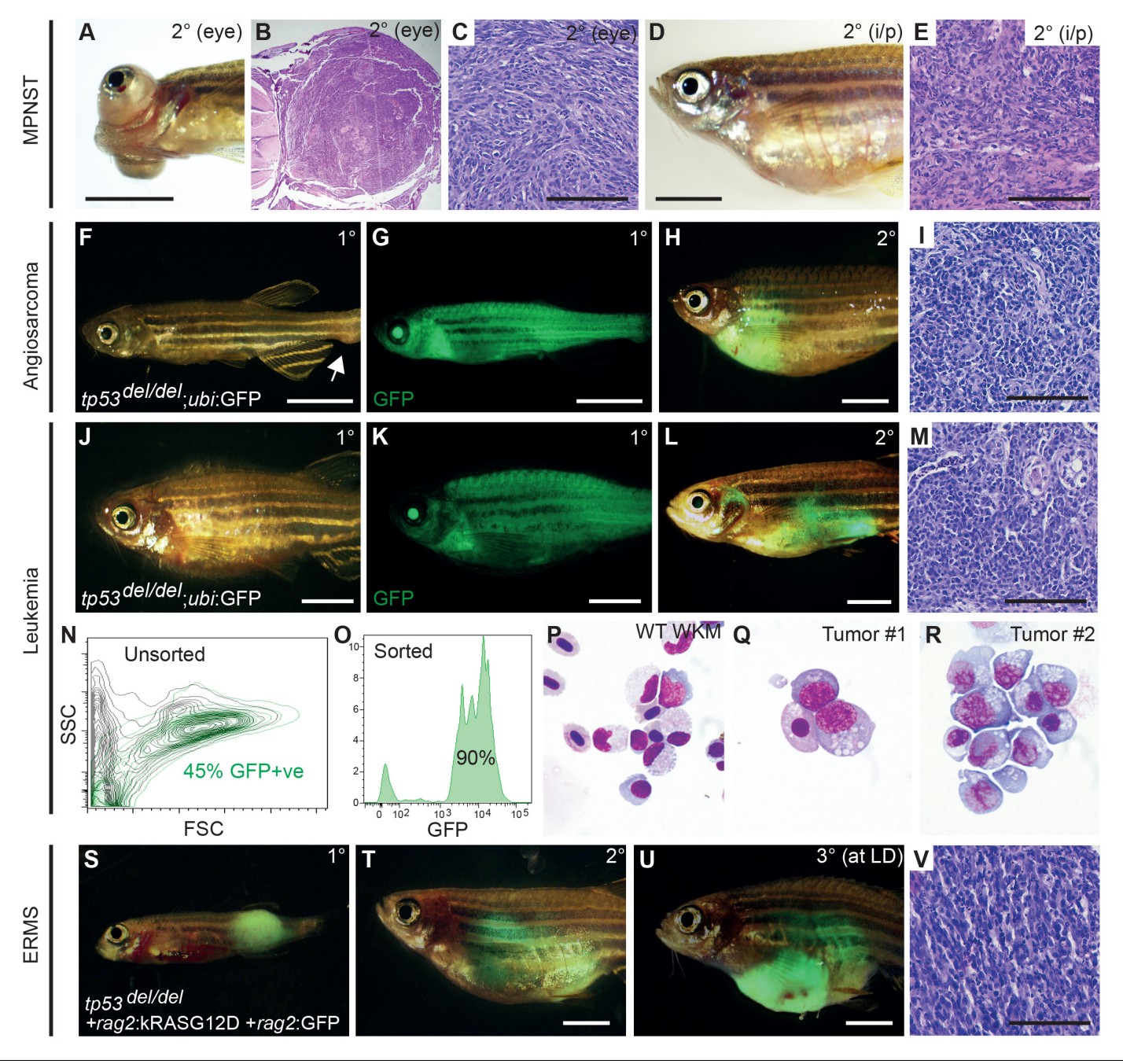

**Figure 2.** *tp53^{del/del}* tumors efficiently transplant into syngeneic CG1 strain zebrafish. (A–E) A primary *tp53^{del/del}* MPNSTs that formed in the eye transplanted orthotopically into the periocular space (A–C) or into the peritoneum of CG1-strain recipient fish (D–E). Intraperitoneal injection (i/p). (F–I) *tp53^{del/del}* Tg(*ubi*:GFP)-positive angiosarcoma. Primary tumor-bearing fish (F–G) and transplanted animal (H–I). (J–R) *tp53^{del/del}* Tg(*ubi*:GFP)-positive leukemia. Primary leukemia (J–K) and transplanted leukemia shown at 20 days post-transplantation (L–R). Whole kidney marrow was isolated from leukemia-engrafted fish and analyzed by FACS (N–O). (N) Forward and side scatter plot of whole kidney marrow of unlabeled CG1 host animal to assess *ubi*:GFP-positive *tp53^{del/del}* leukemia cells following transplantation. (O) Analysis of GFP+ *ubi*:GFP-positive *tp53^{del/del}* leukemia cells following FACS. Purity was ≥90%. (P–R) Cytospins and Wright/Giemsa staining of whole kidney marrow cells isolated from wildtype fish (P) compared with FACS sorted GFP+ cells from two representative aggressive NK cell-like leukemias, showing large blastic cells with abundant basophilic, vacuolated cytoplasm (Q–R). (S–V) Embryonal rhabdomyosarcoma arising in *tp53^{del/del}* fish micro-injected at the one-cell stage with linearized *rag2*:kRAS^{G12D} + *rag2*:GFP. Primary (S), transplanted (2°) (T), and serially transplanted ERMS (3°) (U,V). Whole animal bright-field images (A,D,F,J) and merged GFP-fluorescence images (G,H,K,L,S–U). Hematoxylin and eosin stained sections of engrafted tumors (B–C,E, I, M,V). Scale bars are 5 mm in whole animal images and 100 μm for histology images.

*Figure 2 continued on next page*

*Figure 2 continued*

DOI: https://doi.org/10.7554/eLife.37202.006

The following source data and figure supplement are available for figure 2:

**Figure supplement 1.** Engraftment of $tp53^{del/del}$ tumors into CG1 recipient zebrafish.

DOI: https://doi.org/10.7554/eLife.37202.007

**Figure supplemenrt 1—source data 1.** Source data for *Figure 2—figure supplement 1*.

DOI: https://doi.org/10.7554/eLife.37202.008

identified in zebrafish was significantly enriched for GO terms associated with immune system processes, leukocyte activation, immune response and lymphocyte activation. By contrast, angiosarcomas were enriched in GO gene sets associated with vasculature, blood vessel morphogenesis and cellular proliferation. The germ cell tumor showed enrichment of GO gene sets associated with sexual reproduction and gamete formation. As expected, ERMS shared significant overlap of gene signatures with muscle structure, muscle contraction and muscle development.

To assess similarities between $tp53^{del/del}$ tumors and human, we next assessed if zebrafish tumors express tumor-specific gene signatures identified from human angiosarcoma (*Andersen et al., 2013*), MPNST (*Kolberg et al., 2015*) and ERMS (experimentally determined using GEO: GSE108022) (*Figure 3—figure supplement 1—source data 1*). Using Gene set enrichment analysis (GSEA) (*Mootha et al., 2003*; *Subramanian et al., 2005*), we identified significant enrichment of signatures associated with human angiosarcoma (FDR q-value = 0.001, *Figure 3—figure supplement 1A*), MPNST (FDR q-value = 0.00433526, *Figure 3—figure supplement 1B*), and ERMS (FDR q-value = 0, *Figure 3—figure supplement 1C*) in the corresponding zebrafish $tp53^{del/del}$ tumors but not other tumor types (*Figure 3—figure supplement 1—source data 1*). Taken together, these data reveal conserved gene expression programs associated with both the predicted cells of origin and the corresponding human cancer counterpart.

Given that GSEAsig analysis failed to assign leukemias to a specific lineage, we next assessed if these tumors were enriched for signatures associated with normal blood cell lineages identified previously by our group using single-cell RNA sequencing of the zebrafish marrow (*Tang et al., 2017*). Using these lineage-specific gene sets, we found that $tp53^{del/del}$ leukemias expressed markers indicative of NK and NK-like cells but largely failed to express genes associated with other hematopoietic cell lineages (*Figure 3D* and *Figure 3—figure supplement 1D,E*). To independently confirm our results, we next identified the top 200 most differentially regulated genes in leukemias compared to all other tumor types and assessed if these genes were differentially expressed in each zebrafish blood lineage. Significant enrichment was only observed in NK cells (*Figure 3—figure supplement 1E*, p=0.015, one-sided binomial test), supporting a NK cell origin of $tp53^{del/del}$ leukemias. Importantly, $tp53^{del/del}$ NK cell-like leukemias also expressed well-known genes commonly associated with human NK cells, including *il2ga* and *b, jak3, perforins 2,7,* and *8,* and these genes were highly up-regulated when compared to all other tumor types in our analysis (*Figure 3E*).

In humans, aggressive NK cell leukemias (ANKLs) have a very poor prognosis and often express perforins but lack markers of mature T- and B- cell lineages (*Liang and Graham, 2008*; *Suzuki and Nakamura, 1999*). In human disease, ANKLs are associated with Epstein-Barr virus infection, however, CD3⁻/CD4⁻/CD56⁺/CD13⁻/CD33⁻ leukemias without EBV infection and intact germline configured T-cell receptor and immunoglobulin have been reported (*Liang and Graham, 2008*). Interestingly, both *TP53* point mutations and deletions have been identified in human ANK cell leukemias, suggesting a role for TP53 in pathogenesis of this disease (*Soliman et al., 2014*; *Yagita et al., 2000*). Yet, to date no in vivo animal models of ANKL have been reported precluding direct functional assessment of TP53 loss in eliciting transformation of NK cells.

Given that tumor onset and spectrum differ based on the nature of *Tp53* mutation or deletion in mice, we next compared gene expression between MPNSTs arising in $tp53^{del/del}$ and $tp53^{M214K/M214K}$ mutant zebrafish (*Figure 3—source data 4*). As may be expected, we found significant overlap in expression between homozygous *tp53* deletion and point-mutant MPNSTs when compared to whole fish (p=4e-321 for up-regulated genes and p=5e-182 for down-regulated genes, one-sided Fisher's exact test (*Figure 3—source data 4*), confirming the previously described loss-of-function

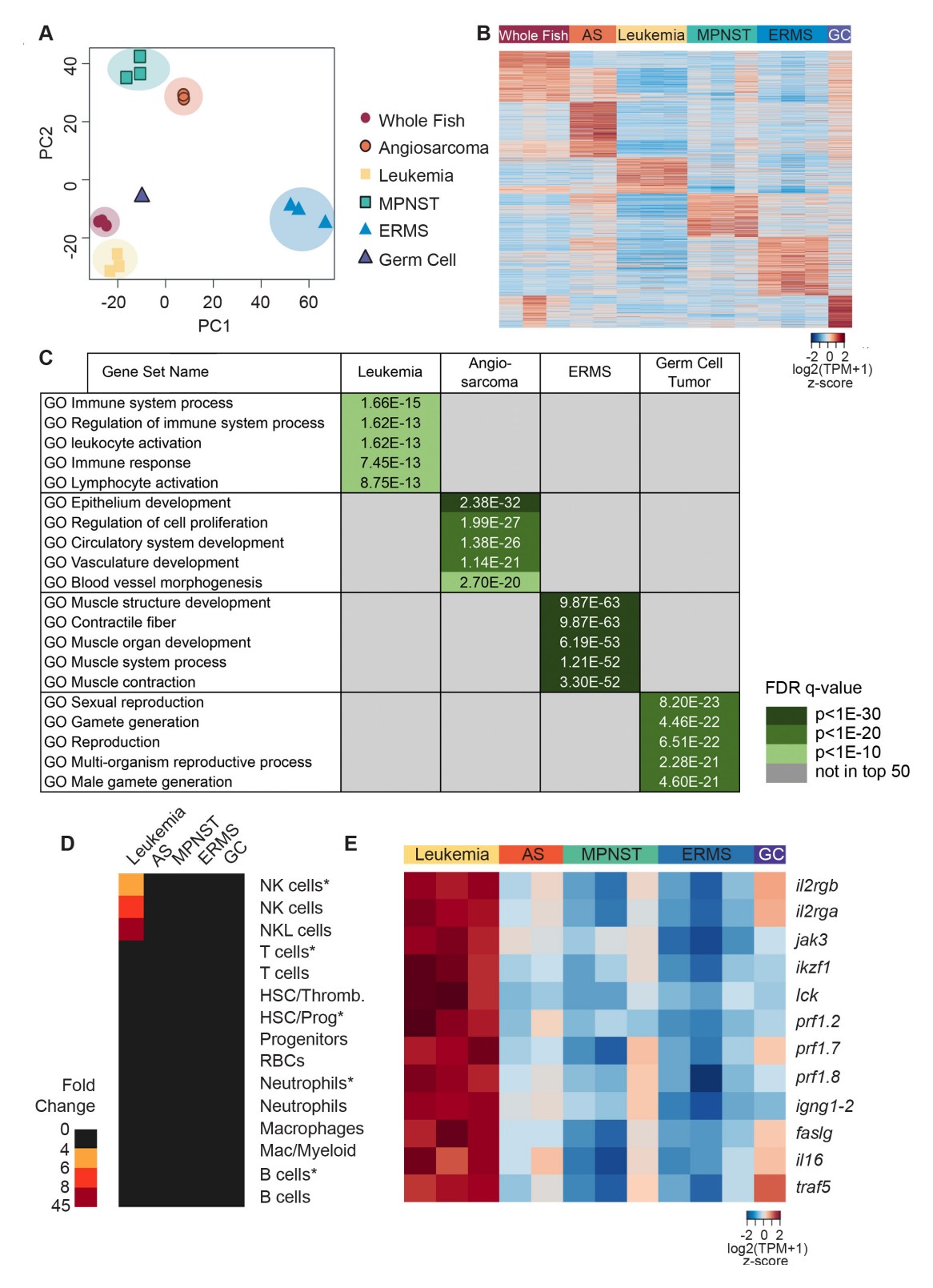

**Figure 3.** Gene expression analysis of tp53$^{del/del}$ tumors. (**A**) Principal component analysis (PCA) of gene expression profiles from whole CG1 syngeneic fish, MPNSTs, a germ cell tumor, and FACS sorted GFP+ leukemia, angiosarcomas, and ERMS. All tumor samples were obtained following engraftment in CG1 syngeneic recipient fish. (**B**) Heat map of genes differentially expressed with respect to controls identifies molecularly defined tumor groups. (**C**) Upregulated genes identified within each tumor type are enriched for Molecular Signature Database (MSigDB) signatures consistent with the expected

*Figure 3 continued on next page*

*Figure 3 continued*

tissue of origin. (D) NK cell leukemias are enriched for gene signatures identified from normal NK and NKL cells in the kidney marrow and NK cells isolated from *rag1*[-/-], *tg(lck:GFP)* transgenic fish (NK cells*). For each analysis, enrichment is shown for the top 30 lineage-restricted genes identified from single-cell transcriptional profiling of transgenic cells using SMARTseq2 (denoted by asterisks) or unsorted cells using InDrops single-cell RNA sequencing approaches. (E) Heat map highlighting NK lineage genes significantly upregulated in *tp53*[del/del] leukemias when compared to all other tumor types analyzed. [log$_2$(fold-change)]. Angiosarcoma (AS) and germ cell tumor (GC).

DOI: https://doi.org/10.7554/eLife.37202.009

The following source data and figure supplements are available for figure 3:

**Source data 1.** Genes and expression values for heatmap rendering shown in *Figure 3B*.

DOI: https://doi.org/10.7554/eLife.37202.014

**Source data 2.** Top 500 transcripts differentially regulated in each tumor subtype identified by RNA sequencing analysis.

DOI: https://doi.org/10.7554/eLife.37202.015

**Source data 3.** Table showing GSEAsig analysis for data rendered in *Figure 3C*.

DOI: https://doi.org/10.7554/eLife.37202.016

**Source data 4.** Differential gene expression for *tp53*[del/del] and *tp53*[M214K/M214K] MPNST.

DOI: https://doi.org/10.7554/eLife.37202.017

**Source data 5.** Genes used for analysis shown in *Figure 3D*.

DOI: https://doi.org/10.7554/eLife.37202.018

**Figure supplement 1.** Zebrafish cancers share common gene expression with human tumors and confirmation of NK-cell linage derivation for *tp53*[del/del] leukemias.

DOI: https://doi.org/10.7554/eLife.37202.010

**Figure supplement 1—source data 1.** GSEA report and human tumor gene expression signatures used for GSEA comparing *tp53*[del/del] angiosarcoma, MPNST and ERMS to their human counterparts.

DOI: https://doi.org/10.7554/eLife.37202.011

**Figure supplement 1—source data 2.** Differential gene expression for *tp53*[del/del] leukemias with respect to blood cells and kidney cells shown in *Figure 3—figure supplement 1D*.

DOI: https://doi.org/10.7554/eLife.37202.012

**Figure supplement 1—source data 3.** Genes used for analysis shown in *Figure 3—figure supplement 1E*.

DOI: https://doi.org/10.7554/eLife.37202.013

activity for *tp53*[M214K] (*Berghmans et al., 2005*). Interestingly, differences in gene expression were also noted when comparing these tumors, likely arising from differences in the underlying mutations

**Table 1.** Results from limiting dilution cell transplantation experiments comparing engraftment potential of *tp53*[wt/wt] and *tp53*[del/del] kRAS[G12D]-induced ERMS.

***tp53*[wt/wt] + *rag2*:kRASG12D ERMS**

| Cell # | Tumor 1 | Tumor 2 | Tumor 3 | Tumor 3 |
|---|---|---|---|---|
| 10000 | 7 of 7 | 5 of 6 | 6 of 6 | 6 of 6 |
| 1000 | 2 of 6 | 2 of 7 | 6 of 8 | 1 of 8 |
| 100 | 0 of 9 | 0 of 8 | 1 of 8 | 0 of 8 |
| TPC# | 1 in 2832 | 1 in 4810 | 1 in 726 | 1 in 7388 |
| | 1 in 3495 (2291–5333) | | | |

***tp53*[del/del] + *rag2*:kRASG12D ERMS**

| Cell # | Tumor 1 | Tumor 2 | Tumor 3 |
|---|---|---|---|
| 10000 | 3 of 5 | 5 of 6 | 6 of 6 |
| 1000 | 3 of 4 | 3 of 5 | 0 of 7 |
| 100 | 3 of 9 | 3 of 7 | 1 of 8 |
| TPC# | 1 in 3546 | 1 in 2228 | 1 in 3640 |
| | 1 in 3038 (1739–5307), p=0.647 | | |

DOI: https://doi.org/10.7554/eLife.37202.019

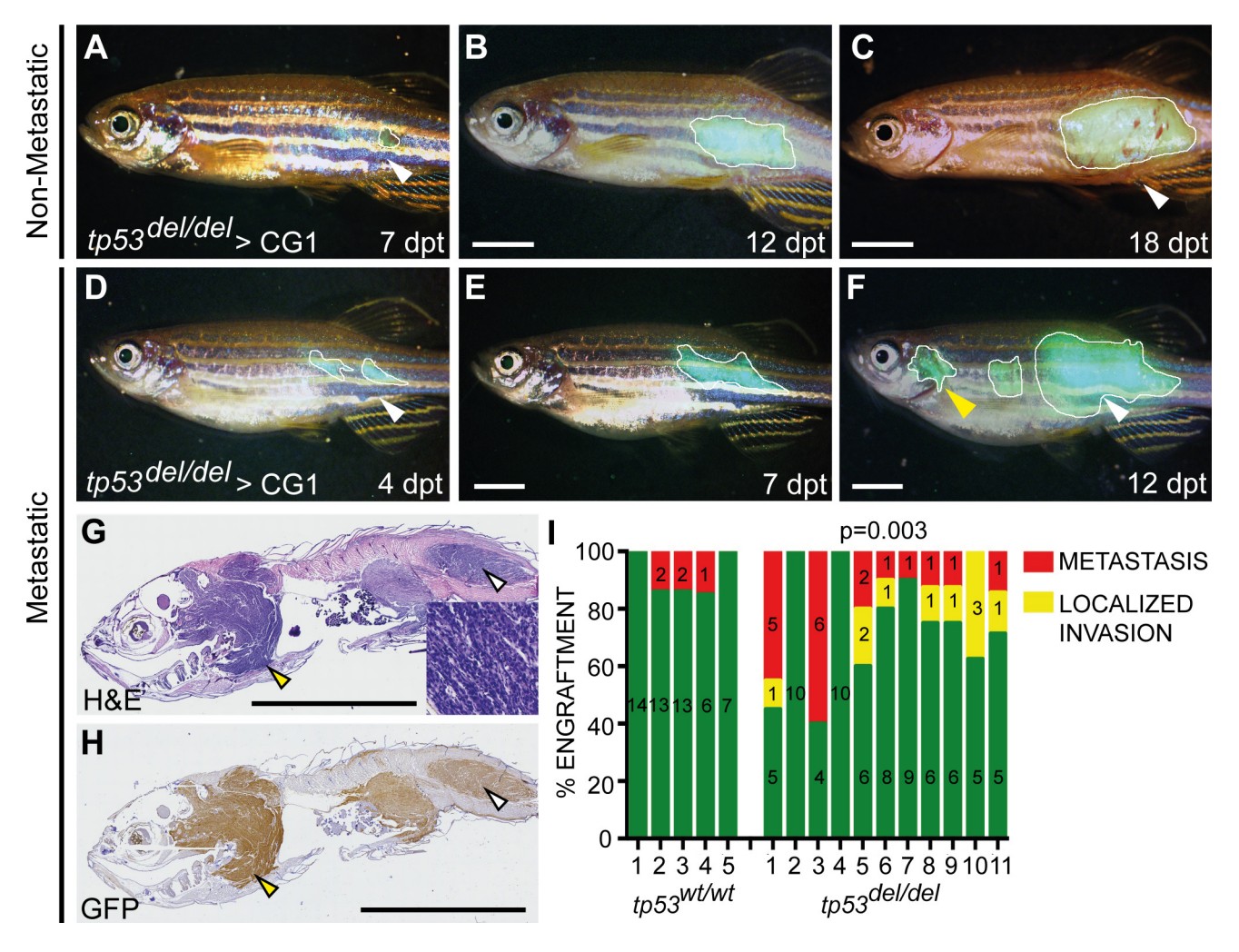

**Figure 4.** *tp53*$^{del/del}$ *kRAS*$^{G12D}$-induced ERMS have increased invasion and metastasis. (**A–F**) Whole animal fluorescent images of CG1-strain fish engrafted into the dorsolateral musculature with non-disseminated (**A–C**) and disseminated ERMS (**D–F**). Days post transplantation (dpt). White lines demarcate GFP+ tumor area. White arrowheads show site of injection and yellow arrowheads denote metastatic lesions. (**G**) H and E and (**H**) GFP immunohistological staining of fish engrafted with metastatic *tp53*$^{del/del}$ *kRAS*$^{G12D}$-induced ERMS. (**I**) Quantification of growth confined to site of injection (green bars) and compared with animals that exhibited local invasion or metastatic ERMS following tumor engraftment until fish were moribund. X-axis identifies 5 *tp53*$^{wt/wt}$ and 11 *tp53*$^{del/del}$ ERMS primary tumors that were transplanted into wild-type CG1 syngeneic host zebrafish. p=0.003, one-sided Fisher's exact test. Scale bars denote 5 mm.

DOI: https://doi.org/10.7554/eLife.37202.020

(*Figure 3—source data 4*), which are known to differentially affect Tp53 function and tumor etiology in genetically engineered mouse models.

## *tp53*$^{del/del}$ ERMS display increased metastasis but did not alter cancer stem cell number

TP53 has roles in regulating self-renewal of normal stem cells and human cancer cells, including acute myeloid leukemia and breast cancer (*Cicalese et al., 2009*; *Meletis et al., 2006*; *TeKippe et al., 2003*; *Zhao et al., 2010*). Thus, we predicted that *tp53* loss may affect the overall frequency of self-renewing cancer stem cells in zebrafish ERMS. To test this hypothesis, GFP+ ERMS cells were isolated by FACS and injected at limiting dilution into the peritoneum of CG1 recipients (1 × 10$^4$–10 cells/recipient, *Figure 2T*). Animals were followed for 90 days for engraftment using whole animal epi-fluorescent imaging. Unexpectedly, kRAS$^{G12D}$-induced tumors harboring wild-type

*tp53* had similar frequency of tumor-propagating stem cells when compared with those of *tp53*^del/del^ ERMS (n ≥ 3 tumors analyzed per genotype, p=0.647 EDLA analysis, *Table 1*). We concluded that Tp53 loss-of-function does not alter the overall frequency of tumor-sustaining, cancer stem cells in ERMS, which contrasts with previous studies that defined major roles for NOTCH1, MYF5/MYOD, and WNT signaling in regulating self-renewal and the overall number of tumor sustaining cell types in rhabdomyosarcoma (*Chen et al., 2014*; *Hayes et al., 2018*; *Ignatius et al., 2017*; *Tenente et al., 2017*).

*TP53* loss is predictive of poor outcome in human ERMS (*Seki et al., 2015*); however, given that *tp53* loss did not regulate the overall frequency of ERMS stem cells in zebrafish, we reasoned that loss of Tp53 might rather affect tumor invasion and metastasis. To test this hypothesis, we undertook tumor cell transplantation experiments whereby GFP-labeled tumor cells were injected into the dorsolateral musculature of recipient fish and animals monitored for spread into the viscera using epifluoresence whole animal imaging (n = 5 *tp53*^wt/wt^ and n = 11 *tp53*^del/del^ ERMS, n = 7–15 recipient fish per tumor,>2×10$^4$ cells/recipient) (*Tang et al., 2016*). As expected, all animals developed GFP+ masses at the site of primary injection (*Figure 4A,D*, n = 160). Tumor growth was followed for up to 30 days after cell transplantation and animals were assessed for 1) local infiltrative disease defined by growth well beyond but contiguous with the primary site, or 2) metastasis defined by growth at sites unconnected to the primary lesion and/or associated with infiltration into organs within the peritoneal cavity (*Tang et al., 2016*). Through serial imaging of engrafted fish over time, we identified only rare metastatic lesions in zebrafish engrafted with *tp53*^wt/wt^ ERMS (n = 5 of 58 engrafted animals, *Figure 4A–F*). By contrast, *tp53*^del/del^ ERMS were highly aggressive and displayed elevated local invasion and disseminated metastatic disease (n = 28 of 102 engrafted fish had metastatic ERMS, p=0.003, one-sided Fisher's exact test, *Figure 4I*). These metastatic lesions were confirmed on paraffin-embedded sections using both hematoxylin/eosin staining and anti-GFP antibody IHC (*Figure 4G,H*). Thus, our in vivo experiments demonstrate an important consequence for *tp53* loss in stimulating local infiltration and metastasis, revealing a property that may account for poor outcome in RMS patients with TP53 pathway deregulation.

Taken together, our work has defined syngeneic zebrafish as a novel model to assess Tp53 loss-of-function phenotypes and has generated a wide array of cancer types now available for study by the community. This is particularly important for modeling angiosarcoma and aggressive NK cell leukemias for which readily available tp53-deficient zebrafish models are lacking. To date, our *tp53*^del/del^ zebrafish is the first description of any animal model of aggressive NK cell-like leukemia, highlighting the importance of Tp53 loss in the genesis of these leukemias and opening exciting new avenues of future study. Finally, our work in embryonal rhabdomyosarcoma revealed that Tp53 loss likely has major impacts on regulating ERMS invasion and metastasis, without altering the overall frequency of relapse driving cancer stem cells. Such findings likely account for why human RMS are more aggressive following TP53 pathway disruption (*Seki et al., 2015*). This work is important, because unlike available genetically engineered mouse models and human ERMS xenografts, zebrafish ERMS are metastatic, which can be readily quantified and visualized in vivo. Future experiments will likely utilize the *tp53*^del/del^ model to study cancer stem cell self-renewal pathways and metastatic progression in a wider array of tumor types including angiosarcoma, MPNSTs, and available transgenic models that require *tp53* loss.

# Methods and materials

**Key resources table**

| Reagent type (species) or resource | Designation | Source or reference | Identifiers | Additional information |
|---|---|---|---|---|
| Genetic reagent (*danio rerio*) | CG1 | *Mizgireuv and Revskoy (2006)* | | |
| Genetic reagent (*danio rerio*) | *rag2*^E450fs/E450fs^ | *Tang et al. (2014)* | | |
| Antibody | anti-tp53 (zebrafish) | Abcam | ab77813, RRID: AB_10864112 | WB 5 ug/mL |

*Continued on next page*

*Continued*

| Reagent type (species) or resource | Designation | Source or reference | Identifiers | Additional information |
|---|---|---|---|---|
| Antibody | anti-Actin | Sigma | A2066, RRID: AB_476693 | WB 1:200 |
| Antibody | anti-mouse HRP | Sigma | NA931, RRID: AB_772210 | WB 1:1000 |
| Antibody | anti-rabbit HRP | Cell Signalling Technology | 7074, RRID: AB2099233 | WB 1:1000 |
| Recombinant DNA construct | *rag2:kRASG12D* | *Langenau et al. (2007)* | | |
| Recombinant DNA construct | *rag2:GFP* | *Langenau et al. (2007)* *Langenau et al. (2008)* | | |
| Recombinant DNA construct | *ubi:GFP* | *Mosimann et al. (2011)* | | |
| Commercial assay or kit | In situ Cell Death Detection Kit, TMR red | Sigma | 12156792910 | |

## Animals

Zebrafish used in this work included: CG1 strain zebrafish (*Mizgireuv and Revskoy, 2006*), CG1-strain Tg(*ubi*:GFP) animals that were generated using Tol2-mediated transgenesis (*Kawakami et al., 2000*; *Mosimann et al., 2011*), and *rag2*$^{E450fs/E450fs}$;*casper* strain zebrafish that were used in a subset of ERMS metastasis assays. All animal studies were approved by the Massachusetts General Hospital Subcommittee on Research Animal Care under the protocol #2011 N-000127.

## Generation of *tp53*$^{del/del}$ zebrafish using TALENS

Four TALEN pairs (two pairs each flanking the *tp53* gene locus in *D. rerio*) were designed to generate a ~12.1 kb deletion encompassing the entire *tp53* coding sequence. Mojo Hand (http://talende-sign.org) was used to design eight 15-mer repeat variable di-residue (RVD) TALENs with a 15 to 18 bp spacer (*Ma et al., 2013*; *Neff et al., 2013*). Each TALEN pair was designed to target a unique restriction site that could be used to determine TALEN cutting efficacy by restriction fragment length polymorphism (RFLP) analysis. All TALEN constructs were synthesized with the Golden Gate method using the RCIscript-GoldyTALEN scaffold (Addgene, https://www.addgene.org/Stephen_Ekker/, ID# 38142) (*Ma et al., 2013*; *Neff et al., 2013*). The RVDs NI, HD, NG and NN (recognizing A, C, T and G bases, respectively) were used to construct TALENs. Intermediate constructs containing RVDs for positions 1 to 10 were synthesized in the pFUS_A receiver plasmid in the first reaction. Pre-synthesized pFUS_B4 plasmids were then selected based on the target sequence. The library of 256 pFUS_B4 plasmids is available through Addgene (https://www.addgene.org/Stephen_Ekker/, Kit # 1000000038). The completed pFUS_A and pFUS_B4 as well as the last half-repeat plasmid (pLR-NI, -HD, -NN or -NG) were combined in the second Golden Gate reaction in the RCIscript-GoldyTALEN expression vector that has T3 promoter. The completed constructs were linearized using SacI, and mRNA was in vitro transcribed using the mMESSAGE mMACHINE T3 Transcription Kit (Thermo Fisher Scientific, cat. no. AM1348). Large deletions encompassing the *tp53* locus were engineered through co-injection of TALEN pairs targeting the *tp53* 5'UTR and 3'UTR. Genotyping was performed using standard PCR: *tp53* forward 5'-CACAGCAAGGACACATCTGC-3', *tp53*$^{del}$ reverse 5'-AGATCAGTGCTTGTATTGTATCAGTTT-3', *tp53*$^{wt}$ reverse 5'-GATCGCTCAGAGTCGCAAA-3'

## Embryonic protein extraction and western blotting

24 hpf embryos of the respective genotypes were dissociated in PBS, spun down at 1000xg to de-yolk samples, and lysed in 10% SDS buffer. Western blot was performed using anti-tp53 (ab77813, Abcam) and anti-actin (A2066, Sigma) antibodies.

## Apoptosis assay

Embryos were raised at 28°C and gamma-irradiated at 24 hpf. At 30 hpf embryos were fixed overnight in 4% paraformaldehyde followed by staining using the In Situ Cell Death Detection Kit, TMR Red (Roche Applied Bioscience) as per manufacturer protocol.

## Histology and immunohistochemistry

Paraffin embedding, sectioning and immunohistochemical analysis of zebrafish sections were performed as previously described (Chen et al., 2014; Ignatius et al., 2012). Anti-human SOX10 was performed at the MGH and BWH DF/HCC Research Pathology Cores. Slides were imaged using a transmitted light Olympus BX41 microscope and a Motic Easy Scan Pro slide scanner. Pathology review and staging were completed by board-certified sarcoma (G.P.N and E.Y.C) and hematology pathologist (R.P.H).

## Micro-injection and ERMS generation

rag2:kRASG12D and rag2:GFP constructs were described previously (Langenau et al., 2008; Langenau et al., 2007). DNA plasmids were linearized with Xho1, phenol:chloroform-extracted, ethanol-precipitated, resuspended in 0.5 × Tris EDTA + 0.1 M KCl, and injected into one-cell CG1 strain embryos.

## FACS and tumor cell transplantation

FACS analysis and RMS cell transplantation were completed essentially as previously described (Chen et al., 2014; Ignatius et al., 2012; Langenau et al., 2007; Smith et al., 2010). tp53$^{del/del}$ angiosarcomas, leukemias and ERMS tumor cells were stained with DAPI to exclude dead cells and sorted twice using a Laser BD FACSAria II Cell Sorter. Sort purity and viability were assessed after two rounds of sorting, exceeding 85% and 90%, respectively. GFP+ ERMS tumors were transplanted at limiting dilution and monitored for tumor engraftment under a fluorescent dissecting microscope from 10 to 90 days post-transplantation. Tumor-propagating cell frequency was quantified using the Extreme Limiting Dilution Analysis software package (http://bioinf.wehi.edu.au/software/elda/). GFP + tumor cells were isolated by FACS from a subset of transplanted fish and RNA isolated for RNA sequencing. Subsets of tumors were fixed in 4% PFA and embedded in paraffin blocks, sectioned and stained with Hematoxylin and Eosin. Sorted GFP+ tp53$^{del/del}$ leukemia cells were spun down onto a cytospin slide and processed by Wright/Giemsa staining.

## RNA sequencing and analysis

Paired-end reads from poly(A)$^+$ RNA-seq were aligned to the GRCz10 reference zebrafish genome with STAR v2.4.0 (Dobin et al., 2013) using GRCz10v85 Ensembl annotations. PCR duplicates were removed with Picard v1.95 [http://broadinstitute.github.io/picard/] and reads aligning to ribosomal RNA were removed with RSeQC (Wang et al., 2012) Gene counts were obtained from reads with an alignment quality of at least 10 using featureCounts (Liao et al., 2014) and transformed to transcript per million (TPM) units. Human orthologues of zebrafish genes were obtained from the Beagle database (Tang et al., 2017; available at: http://chgr.mgh.harvard.edu/genomesrus/). Differential expression analysis was performed with DESeq2 (Love et al., 2014), requiring log2(FC) ≥2 and an FDR < 0.05 was required. Each tumor type was individually compared to the control samples. Clustering of differentially expressed genes used the partitioning around medoids (PAM) method in the cluster R package and the Pearson correlation was used as distance. The number of clusters was optimized with the silhouette function from the same cluster R package.

## Gene set enrichment analysis

Human tumor-specific gene signatures were assessed for enrichment in tp53$^{del/del}$ tumor types using GSEA 3.0. (Mootha et al., 2003; Subramanian et al., 2005). Gene signatures were assessed for anigosarcoma (Andersen et al., 2013), MPNST (Kolberg et al., 2015), and ERMS (GEO:GSE108022). The ERMS signature was defined by genes up-regulated in both human and zebrafish kRAS$^{G12D}$-induced tp53$^{wt/wt}$ ERMS when compared with normal muscle (log2(FC) ≥2). GSEA was completed in comparing individual tumor types to all other tumors using the default parameters and 1000 permutations of the data.

## Comparison of $tp53^{del/del}$ and $tp53^{M214K/M214K}$ MPNST

RNA sequencing data from four $tp53^{M214K/M214K}$ homozygous mutant MPNST samples were processed as described above. Differential expression analysis was performed as described above, comparing $tp53^{del/del}$ and $tp53^{M214K/M214K}$ MPNSTs to whole CG1 controls. Statistical significance was assessed with a one-sided Fisher's exact test on a background of genes that were expressed in at least 4 out of 15 samples.

## Molecular signature database (MSigDB) analysis and leukemia similarities with NK cells

The top 500 most differentially regulated genes within each tumor type were identified and assigned human gene IDs using the Beagle database. These humanized gene lists were then queried for overlaps with molecular signatures from MSigDB. Only the top 50 enriched gene sets were analyzed and representative examples of enriched data sets are shown in *Figure 3C*. For blood cell analysis in *Figure 3D,a* 30 gene signature was defined for each of the major blood cell lineages and then cumulative gene expression analyzed across tumor types as described by *Tang et al. (2017)*. The top 200 genes up-regulated in $tp53^{del/del}$ leukemias compared to all other tumors where assessed relative to gene sets generated using SMARTseq as described in *Tang et al. (2017)*.

## Acknowledgements

We thank Dr. Jim Amatruda for sharing zebrafish MPNST RNAseq sample data.

## Additional information

### Funding

| Funder | Grant reference number | Author |
| --- | --- | --- |
| National Cancer Institute | R24OD016761 | Myron S Ignatius<br>Madeline N Hayes<br>Finola E Moore<br>Qin Tang<br>Sara P Garcia<br>Alexander Jin<br>Ashwin Ramakrishnan<br>Sophia Reeder<br>Gunnlaugur Petur Nielsen<br>Eleanor Y Chen<br>Robert P Hasserjian<br>David M Langenau |
| Alex's Lemonade Stand Foundation for Childhood Cancer | | Myron S Ignatius<br>Madeline N Hayes<br>David M Langenau |
| National Cancer Institute | R00CA175184 | Myron S Ignatius<br>Kunal Baxi<br>Long Wang<br>Yidong Chen |
| National Cancer Center | RR160062 | Myron S Ignatius<br>Kunal Baxi<br>Long Wang |
| National Cancer Institute | R01CA154923 | Myron S Ignatius<br>Madeline N Hayes<br>Finola E Moore<br>Qin Tang<br>Sara P Garcia<br>Alexander Jin<br>Ashwin Ramakrishnan<br>Sophia Reeder<br>Gunnlaugur Petur Nielsen<br>Eleanor Y Chen<br>Robert P Hasserjian<br>David M Langenau |

| National Cancer Institute | U54CA168512 | Myron S Ignatius<br>Madeline N Hayes<br>Finola E Moore<br>Qin Tang<br>Sara P Garcia<br>Alexander Jin<br>Ashwin Ramakrishnan<br>Sophia Reeder<br>Gunnlaugur Petur Nielsen<br>Eleanor Y Chen<br>Robert P Hasserjian<br>David M Langenau |
|---|---|---|
| Rally Foundation | Amanda Riley Foundation Research Fellowship | Madeline N Hayes |
| National Cancer Center | R01CA211734 | Madeline N Hayes<br>Sara P Garcia<br>Alexander Jin<br>David M Langenau |
| National Cancer Center | R01CA215118 | Madeline N Hayes<br>Sara P Garcia<br>Alexander Jin<br>David M Langenau |
| National Cancer Institute | UL1RR024150 | Patrick R Blackburn |
| National Cancer Institute | GM63904 | Stephen C Ekker |
| St. Baldrick's Foundation | | David M Langenau |
| Massachusetts General Hospital | Research Scholars Program | David M Langenau |

The funders had no role in study design, data collection and interpretation, or the decision to submit the work for publication.

## Author contributions

Myron S Ignatius, Conceptualization, Data curation, Formal analysis, Funding acquisition, Investigation, Methodology, Writing—original draft, Writing—review and editing; Madeline N Hayes, Data curation, Formal analysis, Investigation, Writing—original draft, Writing—review and editing; Finola E Moore, Conceptualization, Data curation, Formal analysis, Investigation, Writing—review and editing; Qin Tang, Data curation, Formal analysis, Investigation, Writing—review and editing; Sara P Garcia, Gunnlaugur Petur Nielsen, Robert P Hasserjian, Formal analysis, Writing—review and editing; Patrick R Blackburn, Stephen C Ekker, Resources, Methodology, Writing—review and editing; Kunal Baxi, Formal analysis, Investigation, Methodology, Writing—review and editing; Long Wang, Yidong Chen, Resources; Alexander Jin, Eleanor Y Chen, Formal analysis; Ashwin Ramakrishnan, Investigation; Sophia Reeder, Franck Tirode, Data curation; David M Langenau, Supervision, Funding acquisition, Writing—original draft, Writing—review and editing

## Author ORCIDs

Qin Tang ⓘ http://orcid.org/0000-0002-9487-570X
Patrick R Blackburn ⓘ http://orcid.org/0000-0003-0658-1275
Franck Tirode ⓘ http://orcid.org/0000-0003-4731-7817
Stephen C Ekker ⓘ http://orcid.org/0000-0003-0726-4212
David M Langenau ⓘ http://orcid.org/0000-0001-6664-8318

## Ethics

Animal experimentation: Animal studies were approved by the Massachusetts General Hospital Subcommittee on Research Animal Care under the protocol #2011-N-000127

## Decision letter and Author response

Decision letter https://doi.org/10.7554/eLife.37202.031
Author response https://doi.org/10.7554/eLife.37202.032

# Additional files

## Supplementary files

• Transparent reporting form
DOI: https://doi.org/10.7554/eLife.37202.021

## Data availability

Sequencing data has been deposited in GEO under accession code GSE109581

The following dataset was generated:

| Author(s) | Year | Dataset title | Dataset URL | Database, license, and accessibility information |
|---|---|---|---|---|
| Myron S Ignatius, Madeline N Hayes, David M Langenau | 2018 | tp53 deficiency causes a wide tumor spectrum and increases embryonal rhabdomyosarcoma metastasis in zebrafish | https://www.ncbi.nlm.nih.gov/geo/query/acc.cgi?acc=GSE109581 | Publicly available at the NCBI Gene Expression Omnibus (accession no. GSE109581) |

The following previously published datasets were used:

| Author(s) | Year | Dataset title | Dataset URL | Database, license, and accessibility information |
|---|---|---|---|---|
| Qin Tang, David M Langenau | 2017 | Dissecting hematopoietic and renal cell heterogeneity in adult zebrafish at single cell resolution using RNA sequencing [Smart-seq] | http://www.ncbi.nlm.nih.gov/geo/query/acc.cgi?acc=GSE100911 | Publicly available at the NCBI Gene Expression Omnibus (accession no. GSE100911) |
| Qin Tang, David M Langenau | 2017 | Dissecting hematopoietic and renal cell heterogeneity in adult zebrafish at single cell resolution using RNA sequencing [inDrops] | https://www.ncbi.nlm.nih.gov/geo/query/acc.cgi?acc=GSE100910 | Publicly available at the NCBI Gene Expression Omnibus (accession no. GSE100910) |
| Qin Tang, David M Langenau | 2017 | Dissecting hematopoietic and renal cell heterogeneity in adult zebrafish at single cell resolution using RNA sequencing [bulk RNA-seq] | https://www.ncbi.nlm.nih.gov/geo/query/acc.cgi?acc=GSE100912 | Publicly available at the NCBI Gene Expression Omnibus (accession no. GSE100912) |

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
