## [Decision Letter]

Thank you for submitting your article "*tp53* deficiency causes a wide tumor spectrum and elevates embryonal rhabdomyosarcoma metastasis in zebrafish" for consideration by *eLife*. Your article has been reviewed by three peer reviewers, and the evaluation has been overseen by a Reviewing Editor and Marianne Bronner as the Senior Editor. The following individual involved in review of your submission has agreed to reveal his identity: Yariv Houvras (Reviewer #2).

The reviewers have discussed the reviews with one another and the Reviewing Editor has drafted this decision to help you prepare a revised submission.

Summary:

In this work, the authors have constructed a new *tp53* deficient zebrafish strain in a CG1 syngeneic background. This will enable the study of tumors without the need for immunocompromised status. They then go on to show that this loss correlates with invasive and metastatic potential. This work further opens up the field for delineating how this important tumor suppressor can contribute to a variety of tumor progression models.

Essential revisions:

The major experiments asked for by the reviewers can be grouped into the following two categories, one focused on the point vs. del mutant, and the other on the methods used for gene expression analysis:

1) Comparison of the deletion mutant to the point mutant:a) Since *Tp53* mutation in zebrafish only induces MPNST, does it promote RMS metastasis or not? This would be very interesting to know in order to further understand how *Tp53* contributes to RMS progression.

b) Another interesting question is: Are MPNST gene expression profiles derived from *Tp53* mutant fish similar to those of MPNST from *Tp53^del/del^* fish? This may help us to understand how *P53* mutation and loss-of-function behave in regulating gene transcription.

c) The lack of change in tumor propagating stem cells is interesting in the *tp53^del/del^* ERMS. Is this also the case for *tp53*(point mutant) allele ERMS models? If not, this might help bolster the case that *tp53^del/del^* may be a valuable allele.

d) The increased metastasis observed in the *tp53^del/del^* ERMS model may be due to the increase number of spontaneous tumors in *tp53^del/del^*. This doesn't mean that there isn't an increase in metastasis, but that the result might be conflated with the fact that the animal may be unhealthy due to already existing tumors. Have the authors looked at whether the GFP+ positive metastasized fish also have non GFP+ tumors?

2) The gene expression analyses: a) Assigning NK cell of origin to the leukemias. The authors have used their prior gene expression data from single cell studies and overlap with known markers in human NK cells to make this claim. Are there human gene expression data on NK cell leukemia that further supports this claim? Can they exclude the possibility that the leukemic blasts they identify are myeloid in origin but lack differentiated markers and represent a more primitive myeloid state. Can they examine myeloid markers on cytospin preparations to further evaluate this possibility? The authors' claim would be strengthened if they are able to relate the gene expression to published NK cell gene expression signatures, if possible. The authors should consider modifying the claim that they can identify the lineage with absolute certainty based on the available data.

b) The gene expression studies are elegant, yet I would like to see more detail included in the manuscript with regard to certain details. What controls did the authors use for each tissue type? The conclusion sentence (subsection “Gene expression analysis of *tp53^del/del^*tumors arising in transplant recipient fish”, first paragraph) depends on understanding what the comparison is.

c) The GSEA results for *tp53^del/del^* tumors were not entirely surprising given the tissue origin (i.e. I would expect that the angiosarcoma would have enrichment of vascular genes). The data set would hold a lot more value if the tumor profile for each of the tumors were compared to "normal" tissue of the same type (i.e. leukemia tumor cells compared to normal fish blood cells), rather than whole syngeneic fish. Additionally, have the GO sets been compared to GO sets in human disease, and what is the overlap?

---

## [Author Response]

Essential revisions:The major experiments asked for by the reviewers can be grouped into the following two categories, one focused on the point vs. del mutant, and the other on the methods used for gene expression analysis:1) Comparison of the deletion mutant to the point mutant:a) Since Tp53 mutation in zebrafish only induces MPNST, does it promote RMS metastasis or not? This would be very interesting to know in order to further understand how Tp53 contributes to RMS progression.

We analyzed the small cohort of *tp53* point mutant zebrafish that we maintain in the lab and did not have fish in the correct age class to find MPNSTs (they develop around 14 months of age). Moreover, the point mutant line is not in the CG1 strain background, confounding direct comparison to *tp53^del/del^*tumors. Although we agree these experiments would be interesting, potential impacts of strain differences and lack of animals prevented us from being able to complete these experiments during the allotted 3 month resubmission period.

b) Another interesting question is: Are MPNST gene expression profiles derived from Tp53 mutant fish similar to those of MPNST from Tp53^del/del^ fish? This may help us to understand how P53 mutation and loss-of-function behave in regulating gene transcription.

To address this question, we reached out to members of the zebrafish cancer community and have obtained RNAseq data from four *tp53^M214K/M214K^* MPNSTs and compared expression signatures to *tp53^del/del^*MPNSTs. We identified significant overlap between *p53^M214K/M214K^* and *p53^del/del^* MPNST (Author response image 1, p=4e-321 for up-regulated genes and p=5e-182 for downregulated genes; one-sided Fisher’s exact test), suggesting functional commonalities between the different alleles. This is expected given the loss-of-function phenotypes previously reported for *tp53^M214K/M214K^*(Berghmans et al., 2005). We have added these data to the revised manuscript and included gene lists for these comparisons in Supplementary file 5.

**Author response image 1. respfig1:** Venn diagram depicting overlap between up-regulated and down-regulated genes when comparing homozygous mutant *^tp53M214K/M214K^*and *tp53^del/del^*MPNST to whole adult zebrafish.

c) The lack of change in tumor propagating stem cells is interesting in the tp53^del/del^ ERMS. Is this also the case for tp53(point mutant) allele ERMS models? If not, this might help bolster the case that tp53^del/del^ may be a valuable allele.

Sadly, the *tp53* point mutations have not been generated in the syngeneic CG1 strain zebrafish, obviating our ability to complete these interesting analysis. Syngeneic models are required to accurately assess stem cell frequency following limiting dilution cell transplantation (Smith et al., 2010; Blackburn et al., 2012; Ignatius et al., 2012; Blackburn et al., 2014; Ignatius et al., 2017; Hayes et al., 2018; Garcia et al., 2018). Future studies, which are currently beyond the scope of this work, could generate patient-specific point mutations in syngeneic lines and assess for effects on ERMS stem cell frequency.

d) The increased metastasis observed in the tp53^del/del^ ERMS model may be due to the increase number of spontaneous tumors in tp53^del/del^. This doesn't mean that there isn't an increase in metastasis, but that the result might be conflated with the fact that the animal may be unhealthy due to already existing tumors. Have the authors looked at whether the GFP+ positive metastasized fish also have non GFP+ tumors?

We are sorry that our initial presentation of our experimental design was confusing. Here, GFP-labeled *kRAS^G12D^*-induced ERMS were generated in CG1 *tp53^wt/wt^*and *tp53^del/del^*zebrafish. Primary tumor cells were then harvested from both genotypes and transplanted into wild-type, non-GFP expressing CG1 recipient animals. These recipient fish do not transgenically express GFP endogenously, and thus any GFP mass must be derived from engrafted cells.

To rule out the possibility of other spontaneous tumors arising in CG1 syngeneic transplant fish, we looked at H&E stained sections of recipient animals engrafted with *tp53^del/del^*ERMS and confirmed the presence of only ERMS (n=19). A subset of animals were also assessed by anti-GFP immunostaining on section, confirming that identified tumors were only derived from GFP+ ERMS engrafted cells and did not arise from recipient fish tissues.

2) The gene expression analyses:a) Assigning NK cell of origin to the leukemias. The authors have used their prior gene expression data from single cell studies and overlap with known markers in human NK cells to make this claim. Are there human gene expression data on NK cell leukemia that further supports this claim? The authors' claim would be strengthened if they are able to relate the gene expression to published NK cell gene expression signatures, if possible. The authors should consider modifying the claim that they can identify the lineage with absolute certainty based on the available data.

Unfortunately, expression data sets for human ANKL tumors are not currently available, likely due to the rarity of these tumors.

Yet to directly address this important reviewer comment, we have now completed additional analysis to support the similarity of zebrafish NK cell leukemias with normal NK and NK-like cells from zebrafish. For example, we have expanded the original analysis to show individual gene expression data in Figure 3D (see Figure 3—figure supplement 1D) and updated gene expression analysis in Figure 3E, showing a heat map for expression of well-known NK cell marker genes across tumor types and individuals.

We have also now completed additional experiments to support the assignment of these tumors to the NK cell lineage. Specifically, we identified the top 200 most differentially regulated genes in leukemias compared to all other tumor types and assessed if these genes were differentially expressed within defined blood cell lineages from the zebrafish. We specifically assessed expression using the SMARTseq single cell gene expression dataset from Tang et al., 2017 which included HSC/progenitors isolated as *cd41:*GFP^low^ cells from *tg(cd41:GFP)* transgenic zebrafish, T cells from *tg(lck:GFP)* transgenic zebrafish, NK cells from *rag1-/-, tg(lck:GFP)* transgenic zebrafish, myeloid cells from *tg(mpx:EGFP)* transgenic zebrafish, B cells from marrow-derived *tg(rag2:GFP)* transgenic zebrafish, and HSCs from *tg(runx1^+23^:GFP)* transgenic zebrafish. Significant enrichment was only observed in NK cells (Figure 3—figure supplement 1E, p=0.015, one-sided binomial test), supporting a NK cell origin of *tp53^del/del^* leukemias.

Finally*, tp53^del/del^* NK cell-like leukemias also expressed well-known genes commonly associated with human NK cells, including *il2ga* and *b, jak3, perforins 2, 7,* and *8*, and these genes were highly up-regulated when compared to all other tumor types in our analysis (Figure 3E).

Can they exclude the possibility that the leukemic blasts they identify are myeloid in origin but lack differentiated markers and represent a more primitive myeloid state. Can they examine myeloid markers on cytospin preparations to further evaluate this possibility?

To address this reviewer comment, we have used our previously published single cell expression data (Tang et al., 2017) to show significant enrichment of NK cell gene expression in *tp53^del/del^*leukemias and not other cell lineages (see above and Figure 3 and Figure 3—figure supplement 1).

We have also now included analysis of myeloid genes and confirm that they are not differentially upregulated in leukemias when compared with whole fish (see Figure 3—figure supplement 1D and Supplementary file 7).

b) The gene expression studies are elegant, yet I would like to see more detail included in the manuscript with regard to certain details. What controls did the authors use for each tissue type? The conclusion sentence (subsection “Gene expression analysis of tp53^del/del^ tumors arising in transplant recipient fish”, first paragraph) depends on understanding what the comparison is.

We are sorry for this confusion and have amended the revised manuscript to clarify these comparisons throughout.

c) The GSEA results for tp53^del/del^ tumors were not entirely surprising given the tissue origin (i.e. I would expect that the angiosarcoma would have enrichment of vascular genes). The data set would hold a lot more value if the tumor profile for each of the tumors were compared to "normal" tissue of the same type (i.e. leukemia tumor cells compared to normal fish blood cells), rather than whole syngeneic fish. Additionally, have the GO sets been compared to GO sets in human disease, and what is the overlap?

To validate our assigned tumor designations, we have now assessed if zebrafish tumors express tumor-specific gene signatures identified in human angiosarcoma (Andersen et al., 2013), MPNST (Kolberg et al., 2015) and ERMS (experimentally determined using GEO:GSE108022) (Supplementary file 4). Using Gene set enrichment analysis (GSEA), we report significant enrichment of signatures associated with human angiosarcoma (FDR q-value = 0.001, Figure 3—figure supplement 1A), MPNST (FDR q-value = 0.00433526, Figure 3—figure supplement 1B), and ERMS (FDR q-value = 0, Figure 3—figure supplement 1C) only in the corresponding *tp53^del/del^* zebrafish tumors (Supplementary file 4). Taken together, these data reveal conserved gene expression programs associated with both the predicted cells of origin and corresponding human cancer counterpart.